# Back to the Future: Genetically Encoded Fluorescent Proteins as Inert Tracers of the Intracellular Environment

**DOI:** 10.3390/ijms21114164

**Published:** 2020-06-11

**Authors:** Francesco Cardarelli

**Affiliations:** NEST - Scuola Normale Superiore, Istituto Nanoscienze – CNR (CNR-NANO), Piazza San Silvestro 12, 56127 Pisa, Italy; francesco.cardarelli@sns.it

**Keywords:** GFP, inert tracer, cytoplasm, nucleus, fluorescence, time-resolved methods, diffusion

## Abstract

Over the past decades, the discovery and development of genetically encoded fluorescent proteins (FPs) has brought a revolution into our ability to study biologic phenomena directly within living matter. First, FPs enabled fluorescence-labeling of a variety of molecules of interest to study their localization, interactions and dynamic behavior at various scales—from cells to whole organisms/animals. Then, rationally engineered FP-based sensors facilitated the measurement of physicochemical parameters of living matter—especially at the intracellular level, such as ion concentration, temperature, viscosity, pressure, etc. In addition, FPs were exploited as inert tracers of the intracellular environment in which they are expressed. This oft-neglected role is made possible by two distinctive features of FPs: (*i*) the quite null, unspecific interactions of their characteristic β-barrel structure with the molecular components of the cellular environment; and (*ii*) their compatibility with the use of time-resolved fluorescence-based optical microscopy techniques. This review seeks to highlight the potential of such unique combinations of properties and report on the most significative and original applications (and related advancements of knowledge) produced to date. It is envisioned that the use of FPs as inert tracers of living matter structural organization holds a potential for several lines of further development in the next future, discussed in the last section of the review, which in turn can lead to new breakthroughs in bioimaging.

## 1. Introduction

Progress in human science (and knowledge) is often marked by improvements in the imaging technologies, i.e., fundamental steps forward in our capability to decipher the world at smaller and smaller spatial scales. The first Nobel Prize (in physics, 1901) was awarded to Wilhelm Rontgen for his discovery of X-rays and their astonishing ability to allow the noninvasive viewing of the human skeleton. Over the years, several other imaging techniques and their developers have been honored by Nobel awards, up to one of the last Nobel Prizes in Chemistry—jointly awarded to E. Betzig, S.W. Hell and W.E. Moerner “for the development of super-resolved fluorescence microscopy”, i.e., for pushing the potentialities of optical microscopy down to the nanoscale, where the constituents of life play their functions. The underlying principle behind all these discoveries is that “the more we can see, the more we can investigate”, as also reminded to us by the Nobel laureate Martin Chalfie during his dissertation on the discovery of the green fluorescent protein (GFP) in 2008 [1]. It was precisely this latter discovery that brought a revolution into our ability to “see” selected biologic objects of interest within living matter using the optical properties (i.e., fluorescence) of genetically encoded markers. The development of fluorescent proteins (FPs) started with the discovery and isolation by Shimomura and colleagues of the above-mentioned GFP from the Pacific Ocean jellyfish *Aequorea Victoria* [2]. Yet, the importance of this discovery was fully realized only years later when Douglas Prascher first isolated the cDNA of GFP, postulating on its potential use as fluorescent tracer [3] and then Chalfie and collaborators introduced it into bacteria and *Caenorhabditis elegans* for recombinant protein expression [4]. These contributions paved the way to the use of GFP as a genetically encoded marker of a variety of target molecules (mainly proteins) in cells (for a review see Refs. [5,6]). Concomitantly to its use in cells as marker, GFP optical properties were also optimized by genetic manipulation of its aminoacidic sequence, a research line pioneered and ideally represented by Roger Y. Tsien [7], who shares with Chalfie and Shimomura the 2008 Nobel Prize. In general, a colorful palette of variants with different absorption/emission spectra were engineered by means of direct mutagenesis [8,9] and/or isolation from different cnidarians [10]. GFP and GFP-like proteins were also endowed with functions other than fluorescence. For instance, genetically encoded sensors of the environment have been developed from FPs which enable visualization and quantification of various intracellular physiological events in living cells, tissues, and/or whole organisms [10]. As such, FPs overcome standard organic dyes which are inherently limited by the difficulty to control the cell permeabilization (needed for labeling). For instance, FP-based sensors allowed observing and measuring the dynamic behavior of target proteins, with their functions not only becoming optically visible in living matter, but also being optically activated or deactivated [11,12,13]. FP-based sensors have been engineered to probe changes in the chemical concentrations of specific ions (e.g., hydrogen ion, chloride ion) [14,15,16,17] or other physico-chemical parameters of the cell, such as temperature, pressure and macromolecular crowding (for an extensive review on these latter refer to [18]).

Overall, the design and development of FP-based sensors is leading a major evolution in the bioimaging of single molecules, cells, tissues and whole organisms. Alongside the research efforts in exploiting FPs as fluorescent tags and/or sensors, a continuously growing body of experiments have been focused on the use of FPs as physical probes of the intracellular environment in which they are diffusing and colliding, i.e., the intracellular milieu. This is made possible by the almost null unspecific interactions of the classical FP β-barrel with the other components of the intracellular milieu, such as endogenous proteins, membranes, polysaccharides. This peculiar property—in conjugation with the nanoscopic size and intrinsic fluorescence—makes FPs (GFP above all) ideal inert tracers of the intracellular environment in which they are dispersed. A fundamental requisite is that the fluorescence signal can be acquired at the proper spatial and temporal scale, in such a way that the relevant quantitative information on the environment can be extracted. This is the reason why the interest in FPs as inert tracers of the intracellular environment can be paralleled by the concomitant development of time-resolved fluorescence-based experimental methods and analytical tools. In fact, the higher the spatial and temporal resolution of the approach used, the higher the accuracy (in both the spatial and temporal scales) achievable in the description of dynamic molecular processes. As a matter of fact, the joined development of FPs and time-resolved methods capable to catch FP signals with increasing efficacy produced a paradigm-shift in the way the structural organization of the cell is probed: from “static” imaging of the intracellular determinants of such organization to “dynamic” imaging of an inert tracer that, diffusing and colliding, affords quantitative information on the intracellular landscape in terms of accessibility/connectivity, obstacles/barriers to motion (see Table 1 for a summary on the use of FPs as markers, sensors, inert tracers). This review reports on the most significant contributions to such paradigm shift. For the sake of clearness, the imaginary journey within cells using FPs as inert tracers of the intracellular environment is divided into three sections dedicated to applications within the cytoplasm, applications within the nucleus and future lines of development, respectively. Concerning this latter, in fact, it is envisioned that this new research field, still in its infancy, may have a bright future ahead. Experimental observations on molecular diffusion collected in the interior of cells may influence the way biochemical reactions take place, with possible significant contributions to our understanding of crucial, still obscure phenomena, e.g., the biologic benefit of anomalous transport, the regulation of protein folding/unfolding, intracellular signaling, target search processes and bimolecular reactions kinetics.

## 2. GFPs as Inert Tracers of Cytoplasmic Architecture

The natural condition of living matter is that of nanoscopic entities, the molecules (proteins, ions, nucleic acids, etc.), rapidly diffusing within the intracellular milieu. Molecules are part of a complex 3D environment that we typically depict as ‘aqueous’, together with a mixture of both diffusing/colliding crowding agents and almost immobile structures/compartments. The presence of water as part of the intracellular milieu is among the most obvious facts in cell biology but, at the same time, surprisingly one of the hardest to probe by experiments. Many pioneering experimental efforts have been proposed through the years aimed at deciphering the intracellular organization of water and solutes. They exploited many different techniques, but they share the idea of using inert and diffusing fluorescent tracers of the intracellular environment (see Table 2 for a summary on the most significant contributions). Worthy of note, the first seminal experiments do not exploit FPs and date back to 1986–87: Luby-Phelps and collaborators, using fluorescence recovery after photobleaching (FRAP), studied the diffusion of fluorescein-labeled, size-fractionated dextrans and Ficolls in the cytoplasmic space of living cells as a probe of the physico-chemical properties of cytoplasm [19,20]. Although they could not distinguish among different physical chemical models for the organization of cytoplasm, they demonstrated that the cytoplasm possesses some sort of higher-order intermolecular interactions (structure) not found in simple aqueous protein solutions, even at high concentration. The capability to study the organization of the intracellular environment was significantly boosted by the introduction of genetically encoded FPs, which rapidly became, thanks to the peculiar properties mentioned above, a reference of inert tracers. Credit goes to the pioneering works by Alan Verkman and collaborators on the diffusion of GFP in the cytoplasm and nucleus of live cells, which date back to the nineties [21,22] (Table 2). The main purpose of their studies was to evaluate the suitability of GFP as a probe for rapid solute diffusion in living cells and to measure the apparent viscosity of cell cytoplasm/nucleoplasm from the rates of GFP rotational and translational diffusion. Of note, in the cell cytoplasm, they found a GFP rotational correlation time (measured through anisotropy decay) of about 36 ns, only 1.5-fold slower than that in non-viscous saline solution. There was no evidence for GFP binding to intracellular structures, which would slow apparent GFP rotation. The rapid GFP rotational rate in cytoplasm was deemed consistent with the notion that particle rotation is relatively unhindered when particle size is smaller than the characteristic spacing between obstacles. The rotational data per *se* would suggest an intracellular viscosity close to that of water. By contrast, however, this picture was not confirmed by measurements on GFP translational motion. By using fluorescence recovery after photobleaching (FRAP), Verkman and coworkers found that GFP translational diffusion in cytosol is reduced by 3.2-fold compared to that in water, a behavior similar to that found for dextrans of comparable size. Also, the reduced GFP translation in shrunken cells and of the faster GFP translation in swollen cells was interpreted as consistent with the barrier properties of intracellular obstacles [21,22]. The (apparent) discrepancy between rotational and translational data must be linked to the different temporal and spatial scales probed in the two approaches. Anisotropy decay measurements, in fact, probe the nanoscale (both in terms of time and space), while FRAP-based assays, although potentially fast in terms of temporal resolution, are inherently limited in spatial resolution by the detection spot of the confocal microscope and therefore invariably yield an average over the details of molecular diffusion below at least 200 nm in radial and 600 nm in axial distance [23,24,25,26]. Worthy of mention, the same reasoning (and relative spatial-resolution limitations) apply to those strategies based on ‘photoactivation’/‘photoconversion’ of the probe instead of ‘photobleaching’ [27,28]. Gura Sadovsky and coworkers, for instance, by using Dendra_2_ as a model protein (and inert tracer as well) and photo-converted intensity profile expansion (PIPE) to directly measure its motion and quantify it using an effective diffusion coefficient [27]: the retrieved main conclusions recapitulate what already observed by FRAP (Table 2). In general, a satisfactory explanation of the apparent discrepancy between anisotropy decay measurements in the nanosecond time window and photobleaching-based experiments in the millisecond-to-second time window has remained elusive for decades. Theoretical models, in fact, do potentially provide explanations for suppressed translational motion, pointing to the presence of macromolecular intracellular crowding (for an exhaustive review see [29]). Still, their predictions drastically depend on the nanoscale organization assigned to crowding agents. On the other hand, experimental discrimination between existing theories in the actual intracellular environment is made difficult by the lack of techniques capable to probe macromolecular crowding at the required spatiotemporal scale, the spatiotemporal scale between that of GFP rotation probed by anisotropy and that of GFP translation probed by FRAP. Of note, single particle tracking (SPT) methods could in principle provide an answer below the diffraction limit but tracking small molecules in 3D is technically challenged by their rapid diffusion [30]. In addition, there is a fundamental lack of inert fluorescent probes capable to yield, upon excitation, the number of photons (per time unit) required for proper localization of the probe. In this regard, fluorescence correlation spectroscopy (FCS) represents a valuable alternative based on the statistical analysis of the transit time of many single molecules across the observation volume with no need to dwell on any of them [31]. Not relying on localization, FCS works well with relatively dim fluorescent molecules, such as GFPs—and at relatively high concentration of the probe—such as that typically obtained upon expression of genetically encoded probes. Still, in its classical implementation as a measurement in one point in space (single-point FCS) it would afford local information on molecular diffusion and concentration at a spatial scale again defined by the diffraction-limited observation spot (i.e., 200 nm in radial and 600 nm in axial distance). In fact, GFP translational diffusion parameters retrieved by single-point FCS in the cytoplasm are almost coincident with those obtained by FRAP, highlighting a 3-5-fold-suppressed diffusivity than dilute solutions [32,33] (Table 2). In a heterogeneous environment such as the intracellular medium, spatial sampling is indispensable in addition to temporal sampling in order to understand if the diffusion properties measured locally are representative of the whole environment. Several strategies have been proposed to this aim. For instance, it was demonstrated that molecular diffusion laws can be recovered by fluctuation analysis at various spatial scales by either increasing [34,35,36] or decreasing [37] the laser spot size. This latter effect can be obtained by spatially modulating the fluorescence emission with stimulated emission depletion (STED) methods [38]. STED allows the direct observation of biological phenomena with sub-diffraction spatial resolution and with the high temporal resolution needed to probe molecular diffusion in a 3D environment. STED-based lateral resolutions of ~40 nm allowed probing the nanoscale spatiotemporal dynamics of lipids and proteins in live cell membranes [37,39]. STED-based nanoscopy applied to FPs in the 3D-intracellular environment, however, is challenged by the rapid diffusion of the tracer and from the substantially high background typical of a FP-base measurement: thus, lateral resolutions not higher than ~120 nm were accessible at first [40]. Of note, Lanzano and coworkers recently integrated the STED-FCS method with two analytical approaches, the separation of photons by lifetime tuning (SPLIT) and the fluorescence lifetime correlation spectroscopy (FLCS), to efficiently and simultaneously probe diffusion in a 3D environment at different sub-diffraction scales [41]. SPLIT-FLCS combination allowed probing, from a single FCS experiment, the diffusion from uncorrelated background, and continuously decreasing-in-size observation volumes in the cell cytoplasm, down to the spatial scale of approximately 80 nm, were reached. As a matter of fact, the authors proved the ability of SPLIT-FLCS to generate the GFP diffusion law at each single point and in a relatively short time. Obtained results highlight two interesting features: i) GFP mobility increases by decreasing the volume of the observation spot; and ii) GFP mobility shows a certain degree of heterogeneity within the cell cytoplasm depending on the location of the measurement (Table 2). To tackle the limitation of looking at a single point in space at each measurement, Baum and colleagues measured protein dynamic behavior simultaneously between hundreds of positions by multi-scale fluorescence cross-correlation spectroscopy using a line-illuminating confocal microscope and GFP as an inert tracer [42]. More in detail, to achieve microsecond time resolution on multiple length scales from 0.2 to ~3 μm, the authors extended the principle of FCS measurements at a single point in the sample to several hundred detection volumes positioned within the cell, where each detection volume corresponds to a pixel of an electron multiplying charge-coupled device (EM-CCD) detector array. Cross-correlation analysis of signals from the different detection volumes yields the diffusion coefficients for transport between selected positions along the line in living cells, a concept previously presented by Digman and Gratton as “pair correlation function” analysis [43]. Since this analysis can be done for all combinations of the detection elements at the same time, thousands of cross-correlation functions can be obtained in a single experiment. Simultaneous measurement of the diffusion coefficient on multiple scales allows for reconstructing the environment in which the transport process under study takes place. From these data Baum and colleagues derived a quantitative model of the intracellular architecture that resembles a random obstacle network for diffusing proteins [42]. This topology partitions the cellular content and increases the dwell time of proteins in their local environment. The accessibility of obstacle surfaces depends on protein size. Interestingly, in fact, the mobility of inert GFP monomers (GFP_1_), trimers (GFP_3_,) and pentamers (GFP_5_) was measured in the nucleus and cytoplasm of human U2OS cells (Table 2). Differently sized GFP multimers ‘sense’ distinct structural environments with different surface-to-volume ratios and correlation lengths, i.e., with different resolution. While larger particles diffuse more slowly and become apparently excluded from certain regions, smaller particles can explore their local environment more thoroughly. The conclusions by Baum and collaborators are drawn at a spatial scale defined by the diffraction-limited observation spot. To tackle this limitation, Di Rienzo and collaborators, fused together the concept of raster image correlation spectroscopy (RICS, [44,45]) with its potential application to variable time scales [46], and finally, used the imaging-derived mean squared displacement (iMSD, [47]) as quantitative readout to probe GFP diffusion properties in the 3D-cellular environment with no a priori assumptions on the biologic system [48]. This approach allowed pushing the temporal resolution of image correlation spectroscopy down to 1 microsecond, being able to measure GFP average displacements well below the diffraction limit [48]. More in detail, each scan speed was used as a filter to select the characteristic temporal scale of molecular displacements that significantly contributed to the measured correlation function. Arbitrary µm-sized areas in the cell cytoplasm were selected to perform sequential raster scans at tunable time scales. The hitherto unexplored timescale below 10 microseconds reveals the unobstructed motion of GFP at the nanoscale in the cytoplasm of a living cell [48] (Table 2). In detail, *i*MSD values in the spatial range from 20 to 100 nm match those measured in dilute solutions on the same scale. This accordance suggests that 3D-Brownian motion of an inert globular protein is possible at this spatiotemporal scale in the cell cytoplasm. Fit to the free diffusion model for *t* < 2 × 10^−5^ s yields a GFP diffusion coefficient in cell (D_0_) of 126 ± 3 µm^2^/s at 37 °C, thus implying that cytoplasmic viscosity at this scale is almost equal to that of dilute solutions, and also matching what concluded from local measurements of protein rotation by time-resolved fluorescence anisotropy [21,22]. On the other hand, the *i*MSD plot above 2 x 10^-5^ s shows a clear deviation from the Brownian regime. This deviation implies reduced protein translational motion, again matching what is typically reported in the literature thus far, based on time-resolved measurements operating on larger spatial and/or temporal scales [29]. For instance, well above the diffraction limit an almost 3-fold-slower GFP motion (as compared to dilute solutions) is observed, in keeping with observations obtained by FRAP [22,25]. It is worth noting that the overall phenomenology observed here in the cell cytoplasm was already predicted in silico by Saxton M.J. for the diffusion of a particle in presence of obstacles [49] and experimentally described for lipids and proteins diffusing on the plasma membrane [47,50,51]. The observation of unobstructed Brownian-like motion in the cell cytoplasm, together with in vitro controls on systems that mimic the macromolecular crowding effects of homogeneously diffusing solutes and of excluded volume, suggests a model in which the movement of GFP is regulated by the excluded-volume effect of immobile structures, rather than by freely diffusing solutes. In this regard, it is worth mentioning that Novak et al. demonstrated by simulations that cytoskeleton filaments are unlikely to constitute diffusion barriers sufficient to suppress diffusion of molecules in the cytoplasm (they would need to fill ~90% of space!) while, by contrast, internal membranes (e.g., the ER sheets, mitochondria, vesicles, Golgi apparatus, etc.) appear to be a more likely candidates [52]. In keeping with this idea, selective disruption of the microtubules network by treatment with 10μM Nocodazole did not significantly alter GFP behavior in the cytoplasm [48]. At this point, it is worth noting that the density of crowding agents can be also addressed separately from their diffusion. For instance, a macromolecular crowding FRET sensor was engineered by using FPs by Boersma and collaborators [53]. The protein inserted between a cerulean and a citrine pair formed an artificially designed hinge structure domain composed of two α-helical peptides of A(EAAAK)6A and the flexible linker of (GSG)6. The hinge-like domain displays an open- or condensed conformation depending on the crowded condition due to the excluded-volume effect. The transition between the open and condensed conformation are measured by FRET [53]. In addition, starting from pressure-sensitive YFP, Morikawa and collaborators proposed one glycine insertion which makes YFP sensitive to hydrophobicity in solution, while the wild-type YFP and CFP did not exhibit this sensitivity [54]. The FRET pair of CFP and the glycine-inserted FP yielded a crowding sensor able to detect the changes in protein concentrations during the cell division or upon cell swelling/shrinking [54].

## 3. GFPs as Inert Tracers of Nuclear Architecture

The dominant mode of motion of inert molecules throughout the nucleus is, by a matter of fact, diffusion. How this motion is directed and regulated, however, has several significant physiological implications with potential impact on our understanding of target–search processes taking place in this compartment. In general, the conjugation of time-resolved methods to the use of FPs to study the intranuclear environment mirrors what described above for the cytoplasm. Looking at the literature in time, in fact, a similar advancement in the spatiotemporal accuracy achievable by the different strategies can be found. In 2000, Wachsmuth and collaborators investigated spatial variations of the diffusion behavior of the EGFP (F64 L/S65 T) and of the EGFP-beta-galactosidase fusion protein in living cells by fluorescence correlation spectroscopy [55]. By using a precision x–y translation stage, the authors achieved submicron spatial resolution and a detection volume smaller than a femtoliter.

They found that the diffusional behavior of EGFP deviates from ideal behavior and depends on the position in the cell nucleus. The fluorescence correlation spectroscopy data can either be evaluated as a two component model (with one fraction of the molecules undergoing free Brownian motion with a diffusion coefficient approximately five times smaller than in aqueous solution and another fraction diffusing one or two orders of magnitude slower) or, alternatively, by fitting the data to an anomalous diffusion model where the time dependence of the diffusion serves as a measure for the degree of obstruction, which is large in the nucleus [55]. Possible mechanisms for this long tail behavior include corralling, immobile obstacles, binding with a broad distribution of binding affinities, diffusion within the complex environment defined by the chromatin. In regard to this latter, Dross and coworkers addressed the diffusion of inert fluorescent proteins in living cell nuclei by FCS, but with a two-color confocal scanning detection system [57] (Table 2). The authors show for various human cell lines that the mobility of GFP varies significantly within the cell nucleus but does not correlate with chromatin density (labeled in red). The intranuclear diffusional mobility strongly depends on protein size: in a series of GFP-oligomers, in fact, the diffusion coefficient decreases from the monomer to the tetramer much more than expected for molecules free in aqueous solution. Still, the entire intranuclear chromatin network is accessible to inert proteins up to the size of the GFP-based tetramer, regardless of the chromatin density or cell line. Even the densest chromatin regions do not exclude untagged GFP-based monomers or multimers [57]. The measurements discussed so far, although informative, remain inherently local in space. As discussed above, spatiotemporal correlation spectroscopy holds the potential to tackle this issue. Hinde and collaborators, for instance, addressed the regulation of GFP diffusion within the nucleus by the pCF approach [58]. As mentioned earlier, pCF measures the time a particle takes to go from one location to another by correlating the intensity fluctuations at specific points on a grid independently of how many particles are in the imaging field [43,60]. In such a way, this method provides single-molecule sensitivity in the presence of many molecules and affords a description of the spatial heterogeneity of the intracellular environment in terms of barriers and/or obstacles to diffusion. For instance, if the chromatin network is permeable to GFP diffusion, but to a different extend depending on the spatial location, this will appear as a characteristic feature (gap of communication or delayed communication) in the corresponding pCF carpet. In analogy to previous local FCS measurements, Hinde and collaborators used GFP monomers as inert tracers while Hoechst was used to label chromatin in different channel [58]: interestingly, while standard autocorrelation analysis in the interphase nuclei well agrees with the observation by Dross et al. that the local diffusion coefficient of GFP is independent from DNA density [57], the spatial analysis afforded by pCF highlights the dependence of the diffusive route taken by GFP on DNA density [58]. More in detail, pCF analysis shows GFP migration paths which allow for communication, but also barriers to diffusion resulting in poor or no communication. It appears that GFP diffusion is not continuous throughout the nucleus, where high-density DNA forms a networked channel that allows GFP to diffuse freely throughout, but with restricted ability to cross the channel barriers to the low-density DNA surrounding environment [58] (Table 2). Similar measurements were conducted also throughout the cell cycle, with a particular focus on the mitotic phase [61,62]. It is found that, in this case, the densely packed mitotic chromatin is acting just as a physical barrier to GFP free intranuclear diffusion, allowing delayed, but continuous molecular flow of GFP in and out of a high chromatin density region.

Overall, measurements on inert tracers in the nucleus suggest a model of two virtually disconnected DNA environments, differentiated by density that allow continuous molecular diffusion throughout either and restricted movement in between. The observation of channeled molecular diffusion within the nucleus is compatible with at least two theoretical chromatin models based on simulation of intranuclear diffusion of inert molecules. The first is the phenomenological ICD (interchromosomal domain) model [63], which postulates a network of channels that enable diffusion of small molecules between chromosomal domains. The second is the MLS (multi-loop sub-compartment model) [64], which describes the interphase nucleus as being filled with chromosome territories that intermingle only moderately but allow inter- as well as intrachromosomal transport of small molecules via obstructed diffusion. In keeping with the overall picture emerging for inert-tracer diffusion within the nucleus, the *i*MSD values measured within the intranuclear environment at very short temporal scale yielded a very different trend with respect to the cytoplasm [48]. In particular, GFP intranuclear motion resulted to be not coincident with that in a dilute solution throughout the considered spatiotemporal scale, down to 1 µs [48]. This in turn suggests that excluded-volume effects on protein motion weight much less in the nucleoplasm (i.e., a structurally different compartment, for instance devoid of membranes) than in the cytoplasm, also in keeping with previously reported attempts for mesoscopic modeling of protein diffusion in the various cellular compartments [65]. The persistence of ‘anomalous’ GFP intranuclear diffusion up to the microsecond timescale does not directly contradict rotational diffusion data and concomitantly well agrees with the GFP reduced translational motion typically observed by diffraction-limited measurements [21,22].

## 4. Future Perspectives

The possibility to quantitatively address the regulation imparted by the intracellular environment to the diffusion of molecules has greatly evolved in the last decades of research, mainly thanks to two pillars: i) the development of optical microscopy tools with increased spatial e and temporal resolution; ii) the introduction and adaptation of fluorescent probes as inert tracers of the intracellular environment. Although an extensive review of point (i) is far beyond the scope of the present work, the crucial role of time-resolved fluorescence-based techniques clearly emerged from what discussed. It is undoubted that new technological developments in the near future will further push the spatial and temporal resolution at which molecular processes can be probed in the cell (Figure 1a). Worthy of mention, on one side, new localization strategies based on either the combination of Stimulated Emission Depletion (STED) with electro-optical scanning [66] or the use of local excitation–intensity zeros for localizing emitters [67] have been recently proposed and allowed 3D-localization experiments to reach unprecedented spatial (nanometer) and temporal (millisecond) resolutions. On the other side, new spatiotemporal fluorescence correlation spectroscopy algorithms are continuously increasing our ability to grab quantitative information on molecular motion by statistical analysis of signal fluctuations [68]. In particular, spatial pair-cross-correlation in two dimensions (2D-pCF) was used to obtain relatively high resolution images of molecular diffusion dynamics and transport in live cells using, among others, GFP as inert tracer [69], [59]. The 2D-pCF method measures the time for a particle to go from one location to another by cross-correlating the intensity fluctuations at specific points in an image: hence, diffusion spatial isotropy/anisotropy as well as visual maps of the average path followed by molecules can be created [59,69] (see also Table 2). Finally, continuous technological developments on the optical-microscopy side promise to push all the discussed analytical tools to perform three-dimensional, deeper and faster within living matter (e.g., Hedde and coworkers achieved the rapid measurement of molecular transport and interaction in 3D within living cells using single plane illumination, [56]) (Table 2).

Concerning point (ii) that is, by purpose, the focus of the present review, it was thoroughly discussed how GFP, used as inert tracer, represented an indispensable tool to probe the effect of the environment on molecular diffusion in living cells. This in turn contributed to build a paradigm shift in the way protein intracellular diffusion is described, from the 3D-reduced motion in a continuum fluid of soluble and colliding crowding agents to the unobstructed motion in a fluid of spatially organized, impermeable macromolecular structures, similarly to what already demonstrated on the plasma membrane [51]. For instance, it can be now speculated that the structural organization of the cell cytoplasm (but also of the nucleus) may act as a ‘supramolecular’ regulatory system, directing protein diffusion through spatially defined paths. Several lines of development on this side can be envisioned (Figure 1b-d). First, brighter and more photo-stable variants of the available FPs would be desirable: they may significantly improve the performances of the fluorescence-based methods typically used to monitor molecular dynamic properties. STED microscopy is a useful example to better explain this point. The smallest size of the observation volume achievable depends on the STED beam intensity level and on the signal-to-noise ratio of the measurement. The signal-to-noise ratio of the measurement, in turn, depends on the brightness of the molecule and on the acquisition time [70,71,72]. Therefore, for a given acquisition time, a smaller size of the observation volume can be obtained by (i) increasing the STED beam power and/or by (ii) increasing the brightness of the molecule. Increasing the STED beam power is limited by the potential photo-damage induced on the fluorophores (e.g., photobleaching of EGFP) on the cells. It is this clear that any further increase in molecular brightness will help to improve the overall performances of the method, in terms of a smaller effective observation volume, a reduced acquisition time or a lower level of STED beam illumination intensity. A possible answer to this challenging objective may be envisioned also based on the promising result recently proposed by Dou and collaborators on the de novo design of new fluorescence-activating β-barrels [73]. The authors first showed how, based on considerable symmetry breaking to achieve continuous hydrogen bond connectivity and eliminate backbone strain, it is possible to build ensembles of β-barrel backbone structures with cavity shapes capable to selectively bind and activate the fluorescent molecule DFHBI both in vitro and in E. coli, yeast and mammalian cells. Of note, these new proteins have more than twice less residues and a smaller barrel (e.g., with a width of 1.5 nm, as compared to 2.4 of natural GFP)—although they are not fully genetically encoded fluorophores (i.e., as mentioned, the addition of exogenous ligands is needed). In general, this new platform sets the stage for the design of even more sophisticated ligand-binding structures (e.g., open to bind fluorogenic molecules of different colors and/or optimized for STED imaging) and/or sensors of the environment. In addition, this would in principle represent an ideal platform to develop new inert (and fluorescent) proteins even bigger than natural GFP. In fact, as thoroughly discussed in this review, the possibility to modulate the size of the tracer allows probing different spatial and architectural features of the intracellular environment (Figure 1b). This task, so far, has been addressed by using multimers of GFP, linked together mostly by flexible peptide linkers. However, this configuration naturally produces a “pearl necklace” effect, in which consecutive β-barrels are aligned, with linkers in between. The overall shape of the cargo is thus increasing in a controlled way, but in a preferential direction (i.e., with a major axis and a minor axis). The effect of producing GFP multimers with such a configuration is clearly discernible in FRAP experiments conducted to study nucleocytoplasmic shuttling of molecules across the nuclear pore complex (NPC), which acts a filter. By using GFP and GFP multimers as inert probes of passive diffusion across the NPC, a cutoff limit for this process can be above 80 kDa (molecular weight of the trimer, GFP_3_) [74,75], and thus well beyond the commonly postulated threshold at around 60 kDa [76,77]. This apparent discrepancy can be explained by assuming, as stated above, that GFP multimers adopt an “anisotropic” spatial configuration. It is thus clear that the possibility to expand the inert-tracer size “isotropically” (something that can be envisioned based on the promising results reported by Dou and collaborators, [73]) would tackle this issue in NPC research and, similarly, in other biologic applications.

It is undoubted that, along the way leading to FP development and optimization, a crucial supportive role may be played by in silico prediction/description of FP diffusion and interactions by means of an increasingly accurate choice of FP-specific functional forms and parameters (e.g., of the force field terms) optimized by means new strategies/algorithms (for an exhaustive review refer to [78]) (Figure 1c). In particular, Trovato and coworkers produced a significative step forward by building a model capable of implicitly re-including some effects lost in classical coarse-graining procedures, such as the shape effects (for crowders), hydrodynamics, hydration/dehydration and interaction specificity (for the tracer) [79,80]. The result is an accurate representation of the multiscale dynamics of the tracer (GFP), involving its internal flexibility, specific protein–protein interactions and diffusion, although for the moment simulated only within the prokaryotic cytoplasm of *E. coli* [79]. Although the main conclusions are not immediately transferable to eukaryotic cells, where specific interactions are not negligible and the cytoskeleton/internal membranes are believed to entangle and direct the macromolecule motion, the innovative parameterization proposed is transferable and flexible enough to be applied to different biologic contexts.

A worthy-to-mention additional line of development is the use of GFP-based inert tracers to probe the luminal environment of subcellular structures or organelles (Figure 1d). Solute diffusion within the aqueous lumen of intracellular organelles is involved in many processes, such as metabolism in mitochondria and protein processing in endoplasmic reticulum. In regard to the mitochondrial matrix, theoretical considerations suggest that biochemical events may occur by a “metabolite channeling” mechanism in which metabolites pass from one enzyme to another in an organized complex without aqueous-phase diffusion [81]. Analogously to what done in the cytoplasm, this hypothesis was tested by FRAP and anisotropy-decay measurements using GFP as inert tracer targeted to the mitochondrial matrix of fibroblasts, liver, skeletal muscle and epithelial cell lines [82]. Quantitative analysis of bleach data using a mathematical model of matrix diffusion gave GFP diffusion coefficients only three to fourfold less than that for GFP diffusion in water. Measurement of the rotation of unconjugated GFP by time-resolved anisotropy gave a rotational correlation time similar to that obtained in water. Overall, data suggest that metabolite channeling may not be required to overcome diffusive barriers, while clustering of matrix enzymes in membrane-associated complexes may serve to establish a relatively uncrowded aqueous space in which solutes can freely diffuse [82]. A similar study was carried out to measure solute diffusion in the endoplasmic reticulum [83], highlighting moderately slowed GFP diffusion in the lumen of this compartment. These examples show that GFP diffusion is measurable in intracellular organellar compartments. However, because organelles often have a complex geometry, as well as internal barriers, analytic methods are required to quantitatively interpret the results. Also, experiments are severely challenged by the restless, rapid movement in the 3D-cellular environment of these objects. State-of-the-art optical microscopy tools for delivering subcellular information at molecular resolution, in fact, fail to subtract the 3D-evolution of the entire system while preserving the spatiotemporal resolution required to probe fast dynamic molecular events. It was demonstrated that this bottleneck can be tackled by focusing an excitation light-beam in a periodic around the nanostructure of interest by feedback-based orbital imaging and tracking, a technique first proposed to address the movement of single molecules/particles [84] and then extended to entire subcellular structures/organelles [85,86,87,88,89,90,91]. By this approach, state-of-the-art imaging and analytical tools (e.g., ultrafast spatiotemporal correlation spectroscopy, spectral information, lifetime and polarization) can be used to measure biochemical parameters (e.g., molecular dynamics, interactions and oligomerization state) on moving, nanoscopic, reference systems, where single-molecule sensitivity, high spatiotemporal resolution, and large volume sampling are concomitantly needed in a single measurement. For instance, Begarani and coworkers recently used RICS analysis along the trajectory of a trafficking lysosome (and 6-acetyl-2-dimethylaminonaphthalene, ACDAN, as a lysosome–lumen marker) to probe polarity fluctuations in the lumen of the organelle and their link the organelle metabolism [86]. The use of FPs in such measurements was hampered so far by the need to focus the observation spot on a single fluorescent object (the organelle/structure) of interest for the whole duration of the measurement. This requirement, in fact, is not compatible with the relatively high propensity to photobleaching of standard FPs, unless these latter are able continuously exchange, a very unlikely event in the lumen of an organelle. Regarding this research topic, the possibility to conjugate organic fluorophores to the intrinsic, genetically encoded, inertness of the FP β-barrel (as discussed above based on recent findings by Dou and collaborators, [70]) will represent again an invaluable breakthrough.

In conclusion, it is envisioned that the use of FPs as inert tracers will continue to benefit from new developments in the adjacent field of optical microscopy, as well as from the engineering of new FP variants with improved photostability and tunable size, thus promoting new advancements in our knowledge of the organization of the intracellular environment and its role in regulating dynamic molecular processes.

## Figures and Tables

**Figure 1 ijms-21-04164-f001:**
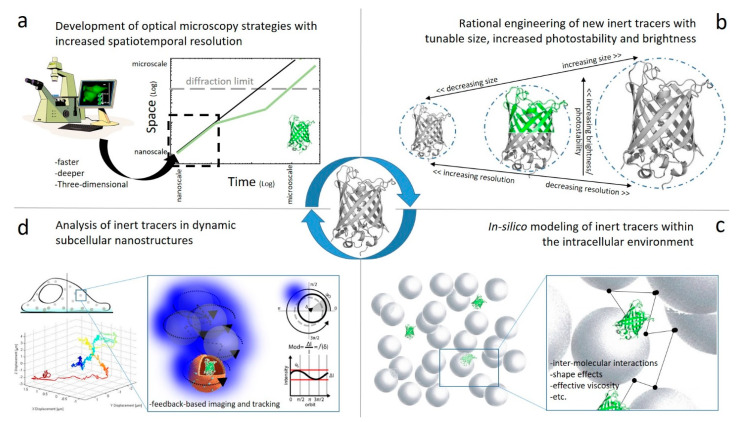
New lines of development concerning the use of genetically encoded FPs as inert tracers of the intracellular environment encompass: (**a**) the development of new fluorescence-based optical microscopy tools with increased spatiotemporal resolution; (**b**) the rational engineering of new inert tracers with tunable size, increased photostability and brightness; (**c**) the effort to model in silico the behavior of inert tracers within the intracellular environment; (**d**) the use of new strategies to grab the behavior of inert tracers in dynamic subcellular nanosystems, such as organelles, transport vesicles, cellular protrusions, etc.

**Table 1 ijms-21-04164-t001:** Genetically encoded fluorescent proteins are mainly used for three major scopes: (i) as fluorescent markers of a molecule of interest, (ii) as biosensors, (iii) as inert tracers of the intracellular environment in conjugation with time-resolved microscopy. Time-resolved microscopy allows to move from a ‘static’ analysis of fluorescent proteins (FPs) localization in space only to a ‘dynamic’ analysis of FP localization both in space and time. This in turn may contain information on the intracellular environment where FPs are diffusing and colliding.

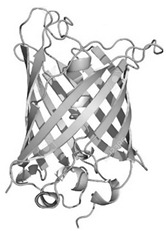	(i) Genetically encoded fluorescent marker of a molecule of interest	See for instance Refs [3,4,5,6,7,8,9,10]
	(ii) Genetically encoded fluorescent biosensor	See for instance Refs [11,12,13,14,15,16,17,18]
	(iii) Genetically encoded fluorescent tracer of the intracellular environment (+ time-resolved fluorescence microscopy) 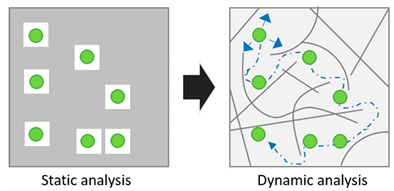	This work

**Table 2 ijms-21-04164-t002:** Summary of the most relevant contributions on the use of genetically encoded FPs as inert tracers of the intracellular environment. The spatial and temporal scale of reference is reported for each selected study, as well as the peculiar FP used, the biologic system chosen and a brief summary of the methodological approach employed, and results obtained.

Spatial Scale (nm)	Temporal Scale (ms)	Inert Tracer	Biologic System	Method/Result	Ref.
>1 µm	milliseconds	GFP	CHO cell cytoplasm	**FRAP:** GFP diffusivity is estimated by fitting the recovery curve. Result: GFP diffusion is 3–5-fold more suppressed than dilute solutions. This is interpreted as the result of macromolecular crowding	Refs. [21,22]
>1 µm	milliseconds	Dendra2	COS7 cell cytoplasm	**PIPE:** GFP diffusivity is estimated by fitting the time expansion of the photoactivated spot. Result: GFP diffusion is 3–5-fold more suppressed than dilute solutions	Ref. [27]
>1 µm	µ-to-milliseconds	GFP	CHO cell cytoplasm	**RICS:** GFP diffusion is extracted from raster-scan images averaging over the entire image. Result: GFP diffusion is 3–5 fold more suppressed than dilute solutions	Ref. [44,45]
>1 µm	milliseconds	GFP	CHO cell cytoplasm	**SPIM-iMSD:** GFP diffusion is measured on a grid of points simultaneously. iMSD analysis yields the average GFP diffusion law within the intracellular environment. Result: GFP diffusion is 3–5-fold more suppressed than dilute solutions	Ref. [56]
200–300 nm	µseconds	GFP	Cell cytoplasm	**Single point FCS:** local measurement of GFP concentration and diffusion. Result: GFP diffusion is still 3–5-fold more suppressed than dilute solutions	Ref. [33]
200–300 nm	µseconds	GFP, GFP multimers	Cell nucleus	**Single point FCS in multiple locations:** GFP diffusion is measured on a grid of points, consecutively. Result: GFP diffusion is suppressed; no correlation is found between GFP diffusivity and chromatin density	Ref. [57]
From 200–300 nm to several microns	Hundreds of µseconds	GFP	CHO cell nucleoplasm	**pCF analysis on a line:** GFP diffusion is measured on a grid of points along a scanned line. Result: cross-correlation of points highlights disconnected GFP flow across chromatin density barriers	Ref. [58]
200 nm	50 µs	GFP, GFP_3_, GFP_5_, RFP	U2OS cell cytoplasm and nucleus	**Multiscale fluorescence cross-correlation spectroscopy:** GFP diffusion is measured on a grid of points simultaneously. Cross-correlation of points is used to measure the GFP transit time to reach the different locations. Result: the regulation imparted by the intracellular structural organization on FPs diffusion is characterized	Ref. [42]
80–100 nm	µseconds	GFP	CHO cell cytoplasm	**Single point STED-FCS:** GFP diffusion is measured, locally, at a sub-diffraction scale. Result: GFP diffusion is on the average only 2-fold more suppressed than dilute solutions, but spatial heterogeneity is highlighted	Ref. [41]
From 100 nm to whole cell	milliseconds	GFP	MB231 cell cytoplasm	**2D-pCF analysis on SPIM-based measurements:** GFP diffusion is measured on a grid of points simultaneously. The 2D-pCF algorithm enables to draw GFP molecular paths across space. Result: a map of the intracellular connectivity/obstacles to diffusion can be drawn	Ref. [59]
From 20–30 nm to several micrometers	1 µs	GFP, GFP_2_	CHO cell cytoplasm and nucleus	**RICS-iMSD at tunable timescales:** GFP displacement is measured by averaging over many microns at tunable time scales. Result: GFP unobstructed (Brownian) motion is observed below 100 nm, anomalous and then suppressed diffusion (3–5-fold) above 100 nm	Ref. [48]
<5 nm	5–50 ns	GFP	CHO cell cytoplasm	**Anisotropy decay:** GFP rotational diffusion is measured at the nanoscale. Result: GFP rotation is almost unhindered in the cell cytoplasm.	Refs. [21,22]

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
