# Peer review of "Back to the Future: Genetically Encoded Fluorescent Proteins as Inert Tracers of the Intracellular Environment"

_ijms, 2020, doi:10.3390/ijms21114164_

Round 1

Reviewer 1 Report

I am a computational chemist interested in the structure and photophysics of fluorescent proteins.  Although I am familiar with fluorescent proteins I have no expertise in their use as inert tracers of the intracellular environment.

Big picture: This is a straight forward review with 90 references. It places all the work in context and summarizes the papers. It is a good aggregation of the research done. The analysis and suggestions for future directions are marginal. Most of the references are a little older, just a couple in the 2016-2020 range. English is clearly not the authors first language and the manuscript requires extensive editorial work.

Suggested changes: Update with new references

Line 48 – Chalfie did not isolate the cDNA of GFP. That was the work of Douglas Prascher, who was also the first person to get the idea of using GFP as a tracer. (Prasher, D. C., Eckenrode, V. K., Ward, W. W., Prendergast, F. G., and Cormier, M. J. (1992) Primary structure of the Aequorea victoria green-fluorescent protein, Gene 111, 229-233.)

Line 54 – here and in many other places add “intracellular” to “environment” too prevent confusion with “environmental chemistry”

Line 64 – I don’t think Table 1 is needed it is all found in the text

Line 100-264  – I think some subheadings or paragraphs would help here.

Line 111 – 117  Should mention that this is pre-GFP work

Line 344  Not sure about the heading

Line 400-409 Important to point out that these are not genetically encoded tracers. They require the addition of exogenous ligands.

Line 413 replace "aminoacidic" with "peptide"

Line 495 -onwards the references are in Italian …..”e”, “pagg” etc.

Author Response

I would like to thank the reviewer for his/her careful reading of the mansuscript and for highlighting a few points that needed clarification/modification. All the raised comments were carefully considered and addressed in the revised version. In detail:

Suggested changes: Update with new references

Line 48 – Chalfie did not isolate the cDNA of GFP. That was the work of Douglas Prascher, who was also the first person to get the idea of using GFP as a tracer. (Prasher, D. C., Eckenrode, V. K., Ward, W. W., Prendergast, F. G., and Cormier, M. J. (1992) Primary structure of the Aequorea victoria green-fluorescent protein, Gene 111, 229-233.)

Thanks for this comment, we modified the citations as correclty suggested by the reviewer

Line 54 – here and in many other places add “intracellular” to “environment” too prevent confusion with “environmental chemistry”

thanks, done

Line 64 – I don’t think Table 1 is needed it is all found in the text

Line 100-264  – I think some subheadings or paragraphs would help here.

These two revisions were not deemed strictly necessary also based on additional reports, but text and tables were carefully revised/modified to improve clarity and readability

Line 111 – 117  Should mention that this is pre-GFP work

done

Line 344  Not sure about the heading

Thanks, modified

Line 400-409 Important to point out that these are not genetically encoded tracers. They require the addition of exogenous ligands.

the sentence has been modified accordingly

Line 413 replace "aminoacidic" with "peptide"

done

Line 495 -onwards the references are in Italian …..”e”, “pagg” etc.

Modified

Reviewer 2 Report

The author of this Review does a really good job highlighting the most important advances and applications of fluorescent proteins as inert tracers of the intracellular environment.

The Review is insightful and well constructed.

Only one minor comment: the author starts by highlighting the introduction of fluorescent proteins to the field of biological sciences by mentioning only Chalfie and Shimomura. I personally feel he must also recognize the contributions of Roger Tsien, who also shared the Nobel Prize with them and devoted his life to the engineering and use of fluorescent proteins in biology. This would also be an important sign of respect in his regards, given he passed away not so long ago.

Author Response

I would like to thank the Reviewer for his/her overall appreciation of this work and for suggesting to properly cite the contribution by Rogier Y. Tsien. This was done in the revised version.

Reviewer 3 Report

This paper by Francesco Cardarelli constitutes a very interesting review article about the most significant and original applications of genetically-encoded fluorescent proteins as inert tracers of the intracellular environment. The review seems to be very useful and includes a lot of sufficiently detailed and properly ordered information – it is demonstrated by a considerable list of references. The paper is generally properly written and the quality of the text is good. This is important, especially for review articles, to make sure the manuscript is more easily readable by target audience. The references appear to be uniformly formatted, what demonstrates clearly that the list and the whole manuscript were prepared with care. In the Reviewer’s opinion it was a very good idea to make “a list” of the most relevant contributions on the use of genetically-encoded FPs as a table. Such a tabular presentation is very useful and allows for an easy comparison. In my opinion the manuscript merits to be published in International Journal of Molecular Sciences.

Some minor points: in line 11 – remove “to”; remove the text form lines 26 and 27.

Author Response

I would like  to thank the Reviewer for his/her overall appreciation of this manuscript and for highlighting a few minor points to correct. These were carefully addressed in the revised version